# The Influence of Lifestyle on High-Density Lipoprotein Concentration among Mexican Emerging Adults

**DOI:** 10.3390/nu15214568

**Published:** 2023-10-27

**Authors:** Karla Paulina Luna-Castillo, Andres López-Quintero, Lucrecia Carrera-Quintanar, Iris Monserrat Llamas-Covarrubias, José Francisco Muñoz-Valle, Fabiola Márquez-Sandoval

**Affiliations:** 1Doctorado en Ciencias de la Nutrición Traslacional (DCNT), Departamento de Clínicas de la Reproducción Humana, Crecimiento y Desarrollo Infantil, Centro Universitario de Ciencias de la Salud (CUCS), Universidad de Guadalajara (UdeG), Guadalajara 44340, Jalisco, Mexico; karla.luna3337@alumnos.udg.mx (K.P.L.-C.); lucrecia.carrera@academicos.udg.mx (L.C.-Q.); iris.llamas@academicos.udg.mx (I.M.L.-C.); drjosefranciscomv@cucs.udg.mx (J.F.M.-V.); 2Instituto de Nutrigenética y Nutrigenómica Traslacional (INNUGET), Centro Universitario de Ciencias de la Salud (CUCS), Universidad de Guadalajara (UdeG), Guadalajara 44340, Jalisco, Mexico; 3Instituto de Investigación en Cáncer en la Infancia y Adolescencia (INICIA), Centro Universitario de Ciencias de la Salud (CUCS), Universidad de Guadalajara (UdeG), Guadalajara 44340, Jalisco, Mexico; 4Instituto de Investigación en Ciencias Biomédicas (IICB), Centro Universitario de Ciencias de la Salud (CUCS), Universidad de Guadalajara (UdeG), Guadalajara 44340, Jalisco, Mexico

**Keywords:** prevalence of inadequate intake, NOVA system, HDL-C concentrations, emerging adulthood, diet quality

## Abstract

Previous studies have highlighted the role of lifestyle on HDL-C concentrations in adults. To our knowledge, the health and nutritional status of emerging adults have been understudied. The present study aimed to explore the most important lifestyle factors, including micronutrient intake adequacy and the percentage of energy from food processing, according to HDL-C concentrations in emerging adults. In this context, a cross-sectional analysis was conducted on 261 Mexican emerging adults who were apparently healthy. Lifestyle factors were collected through a structured survey and the prevalence of micronutrient intake inadequacy was estimated using the estimated average requirement cut-point method. The percentage of energy from ultra-processed foods was assessed using the NOVA system. HDL-C was determined using the enzymatic colorimetric method. Statistical analyses were conducted in SPSS. The results revealed that lifestyle factors do not differ according to HDL-C status. The participants showed a poor nutritional diet that was energy-dense and micronutrient-inadequate. Nearly half of their energy came from processed and ultra-processed foods. Most participants did not meet the recommendations for key nutrients (ϖ3 fatty acids and phytosterols) that promote a healthy lipid status. In conclusion, regardless of their HDL-C levels, emerging adults exhibited lifestyle-related risk factors. The persistence of these findings over time could contribute to the development of metabolic disorders in the future. It is crucial to increase understanding and to develop effective nutritional interventions during this critical phase of life.

## 1. Introduction

Emerging adulthood (EA) is considered a transition between adolescence and adulthood, typically comprising the ages between 18 and 30 years [1,2]. Until recent years, it was considered a stage of low health risk. However, the constant changes in social, family, and personal contexts, as well as the lack of economic independence [3,4], could represent a critical period for the adoption of lifelong risky lifestyles and eating behaviors [5,6]. During this period, different metabolic alterations may develop and go unnoticed for a long time until the symptoms of pathologies are evident. Previous reports have shown the presence of low HDL-C in university students [7,8,9]. This lipoprotein plays an important role in the prevention of cardiovascular diseases and can be altered by different factors. Lifestyle changes and specific dietary components have been recognized in the modulation of HDL. The most relevant HDL-related components that are already known to affect HDL-C concentrations (5 to 30%) are physical activity (PA), weight loss, smoking cessation, moderate alcohol consumption, energy restriction, dietary lipids (monounsaturated fatty acids, polyunsaturated fatty acids, saturated fatty acids, trans fatty acids), carbohydrates, and antioxidant intake [10,11,12,13,14,15,16,17,18,19,20,21]. PA is considered one the most important strategies with an effect on HDL-C levels. It was widely demonstrated that regular aerobic exercise (at least 150 min/week) or resistance exercise for 15–20 min/week, and even leisure time and occupational PA according to PA guideline recommendations, can increase HDL-C concentrations [19,22,23,24,25,26]. Weight loss resulting from exercise [27] or caloric restriction [28] could be a good method to contribute to increased HDL-C [19]. Other lifestyle components such as cigarette smoking produce a decrease in HDL-C concentrations [29], and for its part, moderate alcohol consumption has been associated with an increase in HDL-C [30,31] and a slower decline over time [31]. Regarding dietary components, low- and very- low-carbohydrate diets [20] as well as their replacement with mono- or polyunsaturated fat have been associated with an increase in HDL-C [14,16,19,32]. In a meta-analysis, high-monounsaturated-fat diets compared with high-carbohydrate diets had a significant HDL improvement [14]. Even when comparing low-carbohydrate diets with low-fat diets, it has a greater HDL-C profile [17]. In addition, recent evidence suggests that the consumption of ultra-processed foods (UPFs) has been related with nutritionally poor diets and the development of chronic diseases [33,34]. UPF consumption has been associated with changes in lipid concentrations (increases in total cholesterol and LDL-C) in children [35]. It has been previously reported that there are unfavorable effects of trans fatty acids resulting from industrial processing on HDL-C levels [11]. Evidence of lifestyle influences on HDL-C, specifically in emerging adults, is limited and unclear. Little attention has been paid to the health and nutritional status of emerging adults. Furthermore, a limited number of studies have described in detail the health and nutrition of this population. To our knowledge, the existing evidence on the effect of lifestyle, UPF intake, and adequate nutrient intake according to recommendations for dyslipidemia and cardiovascular disease prevention on the modulation of HDL-C concentrations focuses on a broad age range that includes adults and older adults (18–80 years) or children, who have entirely different physical, biological, and behavioral characteristics. In this regard, our aim was to explore lifestyle factors such as smoking habits, alcohol consumption, physical activity, and diet characteristics as prevalence of nutrient adequacy and UPF consumption according to HDL-C concentrations among emerging adults.

## 2. Materials and Methods

A cross-sectional analysis of the data and blood samples of participants was included from the multicenter study LATIN America METabolic Syndrome Mexico—LATINMETS [36]. The study protocol was carried out according to the Declaration of Helsinki Guidelines and was approved by the ethics, research, and biosafety committees of the University Center of Health Sciences from the University of Guadalajara (CI:01-921). All participants provided written informed consent.

### 2.1. Study Population

A total of 261 data and blood samples from emerging adults were included: 191 (73.1%) females and 70 (26.9%) males. The eligibility criteria included men and women aged 18 to 30 years, university students in their last semesters of health sciences, or recent graduates, in addition to clinical records with complete information on the sociodemographic, anthropometric, clinical, and biochemical data as well as a complete semiquantitative food frequency questionnaire (SFFQ) of the subjects.

Data for sociodemographic information such as age, sex, hereditary family history, and medication were collected through interviews. Lifestyle characteristics, including physical activity (PA), smoking status, and alcohol use, were collected from a structured survey. PA was assessed using a validated Spanish version of the Minnesota Leisure Time Physical Activity Questionnaire [37,38]. The amount of time spent weekly on PA was calculated based on the frequency of each activity and the average time spent per day. The weekly time spent on PA was categorized according to the World Health Organization recommendations (>150 min per week or <150 min per week). Alcohol intake was calculated from a validated SFFQ [39].

### 2.2. Dietary Assessment

Dietary intake was assessed through an SFFQ, validated on a Mexican population [39]. The participants reported their habitual intake frequency and amount of each food over the previous year. The amount of each food and beverage was calculated by multiplying the standard serving size and the intake frequency of each item (g, mL), divided by seven days, in order to obtain the grams and mL ingested of each food or beverage per day, per subject [40]. Energy, macro- and micronutrients were analyzed using Axxya Systems’ Nutritionist Pro Diet software version 7.9 Standalone (Axxya Systems, Stafford, TX, USA). The dietary ω6/ω3 ratio was evaluated based on the difference between the total dietary ω6 (g) and ω3 (g) intake. The total intake of ω6 was obtained based on the sum of linoleic acid and arachidonic acid, and the total ω3 intake was obtained based on the amount of *α*-linolenic acid (ALA), eicosapentaenoic acid (EPA), and docosahexaenoic acid (DHA).

#### 2.2.1. Prevalence of Inadequate Nutrient Intake

The assessment of energy adequacy was determined based on the proportion of the subjects below who were within and above the desirable range of body mass index (BMI). The nutritional adequacy of macronutrients was compared with the nutritional recommendations, based on the international guidelines for the management of dyslipidemia [10,11,41]. We evaluated the proportion of the group that was outside the range of the percentage of energy from macronutrients. The prevalence of micronutrient intake inadequacy was estimated using the estimated average requirement (EAR) cut-point method, which calculated the proportion of the population who displayed inadequate nutrient intake. Inadequate intake was defined as any micronutrient intake below the EAR if available, or the adequate intake (AI) value from the Institute of Medicine (IOM) [42]. We assessed the following micronutrient intake adequacy: vitamins A, D, E, K, B1, B2, B3, B5, B6, and B12.

#### 2.2.2. Percentage of Energy from Processed and Ultra-Processed Food

The consumption and percentage of energy from processed and ultra-processed foods was assessed using the food classification system NOVA by Monteiro [34], which classified food into four groups: (1) unprocessed (NP) or minimally processed food (MPF), (2) processed culinary ingredients (PCI), (3) processed food (PF), and (4) ultra-processed food (UPF). All food and beverages reported in the SFFQ were classified into the four NOVA food groups. We considered all the groups for the description of this study. The percentage of total energy intake from PF and UPF was calculated by dividing the total energy from UPF or PF consumed (kcal) into total energy consumption per day, multiplied by 100.

### 2.3. Biochemical Analysis

Lipid concentrations, such as total cholesterol, HDL-C, LDL-C, and triglycerides, were determined using the enzymatic colorimetric method. Low HDL-C was defined as HDL-C < 40 mg/dL in men and <50 mg/dL in women, according to the American Heart Association [43,44].

### 2.4. Statistical Analysis

Statistical analysis was performed in IBM SPSS^®^ Statistics 25.0 IOS and GraphPad Prism 9 for Mac. The distribution of continuous variables in normality were analyzed using the Kolmogorov–Smirnov test, a histogram, and kurtosis. Continuous variables were treated with parametric tests because they came from a sufficiently large sample [45,46]. The differences among participants in lifestyle characteristics, percentages of nutrient intake adequacy, and percentages of energy from PF and UPF between healthy HDL-C and low HDL-C were compared using Student’s *t*-test for continuous variables and a chi-square test for categorical variables. A *p*-value < 0.05 was considered statistically significant.

## 3. Results

In this study, 261 emerging adults between 18 and 30 years of age were analyzed. The majority were women (72%) and undergraduate students from different health science careers (69.3%). The participants had a mean age of 22.7 years. Significant differences were found between gender according to their HDL-C status (*p* < 0.05) (Table 1).

The prevalence of low HDL-C in emerging adults was about 26.8% (n = 70), with a mean concentration of 53.5 mg/dL. Significant differences were found between HDL-C status (healthy or low concentration) and total cholesterol, as we expected. The rest of the lipids evaluated were similar between the groups (Table 2).

The presence of previously reported lifestyle characteristics related with HDL-C concentrations were compared between groups (Table 3). The majority practiced >150 min of PA per week. Lifestyle factors related to HDL-C concentration, such as smoking habits and alcoholism, were excessive in around 10% of the total population. Almost 50% of the emerging adults reported weight loss in the last 3 months, which was significantly greater in the low HDL-C group compared with the healthy HDL-C group.

Emerging adults had high PF and UPF intakes; nearly 50% of their energy came from these products. No significant differences were observed in the lifestyle aspects between groups (Table 3).

On the other hand, the description of dietary characteristics focusing on HDL-C-related nutrients and the meeting of energy and macronutrient requirements are presented in Table 4. A poor nutritional diet was observed in the emerging adults. About one third of all the participants had an excessive energy intake (28.7%). The percentage of energy provided from carbohydrates and total lipids were above that suggested in 29.5 and 39.8% of the subjects, respectively. According to the type of fatty acids, most of the emerging adults had an excessive intake of saturated fatty acids (SFAs) (87%), while 40% of the participants had intakes below the recommended intake of omega 6 (linoleic acid). The prevalence of an inadequate intake of the main components of ω3 (ALA, EPA, and DHA) is shown in Table 4, where it is highlighted that 88% of the subjects with healthy HDL and low HDL presented intakes below the recommended levels of eicosapentaenoic acid (EPA) and docosahexaenoic acid (DHA). Close to half of the healthy HDL-C group (46.8%) presented intakes below the range of alpha linolenic acid (ALA) and more than half of the low HDL-C group. The mean ω6/ω3 ratio was 9/1 in the whole population, with a slightly higher ratio in participants with low HDL-C (no significant differences). The total cohort of participants with low HDL-C exceeded the ω6/ω3 ratio recommendation. Together, the low HDL-C and healthy HDL-C groups had a low proportion of the above-recommended trans fatty acid (TFA) intakes (0.5% and 1.4% of participants, respectively). No differences were observed in the mean nutrient intake adequacy of macronutrients when the HDL status was compared (Table 4).

The prevalence of inadequate intake for each vitamin according to low HDL and healthy HDL is shown in Table 5. The average number of vitamins with intakes below the EAR was comparable between groups. The vitamins with the highest prevalence of inadequacy or non-compliance with requirements among the participants were as follows: vitamin E (100%), vitamin B7 (85.1%), vitamin K (58.2%), vitamin B5 (44.1%), and vitamin D (36.4%) in both groups. However, the observed mean intakes below the AI for vitamin E, K, and D can only be inferred to be below the requirement, rather than showing a prevalence of inadequacy.

None of the participants reached the suggested daily intake of phytosterols of 2 g per day. The total carotenoid intake was 13.8 ± 63.1 mg/dL, of which 38.5% were β-carotene (4721.9 ± 2689.8), 32.9% lycopene (4032.0 ± 2099.5 mcg), 14.6% Lutein and Zeaxanthin (1786.6 ± 807.6 mcg), 11.4% α-carotene (1393.2 ± 972.5 mcg), and 2.6% β-cryptoxanthin (324.9 ± 255.0 mcg). A higher mean intake of total carotenoids, lutein, zeaxanthin, β-cryptoxanthin, and lycopene can be observed in participants with a healthy HDL-C concentration compared to those with a low HDL-C concentration; however, there is no statistical significance (Table 6).

## 4. Discussion

One of the main therapies for dyslipidemia and cardiovascular disease prevention and treatment involves lifestyle changes. To the best of our knowledge, this is the first paper that describes and explores the presence of modifiable lifestyle factors according to HDL-C concentrations in emerging adults. In this cross-sectional study, sociodemographic characteristics, PA, smoking habits, alcohol consumption, food processing (PF and UPF intakes), and the prevalence of inadequate nutrient intake (macronutrients, vitamins, and antioxidants) were evaluated in Mexican emerging adults. These elements were chosen according to previous precedents about their effect on the maintenance of healthy HDL-C levels in mature adults. It is unknown whether these effects are relevant or are conserved in emerging adulthood; according to the authors Arnett et al. [1,3], the physical, biological, and behavioral characteristics of adults are different from decade to decade. There is no consistent evidence for the impact of lifestyle factors on HDL-C concentrations at early ages [19,20,21,49,50,51,52].

The results from our study highlighted that, despite there being no differences in lifestyle and nutrient intake according to HDL-C concentrations, there were observed lifestyle-related risk factors for cardiovascular disease at this time. We detected the presence of energy-dense and nutrient-poor diets, supported by the high prevalence of nutrient insufficiency and the high percentage of energy from PFs and UPFs in Mexican emerging adults.

Previously, it has been demonstrated that regular aerobic exercise (at least 150 min/week) or resistance exercise for 15–20 min/week improves HDL-C concentrations [22,23,53]. We observed that most of the emerging adults practiced regular PA regardless of their serum HDL-C concentrations. Given recent reports, moderate to vigorous PA declines throughout life, meaning that the highest PA occurs in early adulthood and decreases in late adulthood [54], which makes sense with our findings. We did not find any differences in PA between subjects with low HDL-C and healthy HDL-C. In line with previous findings on Mexican [55], U.S. [56], and Lithuanian [57] young adults (79.2, 85, and 64%), we observed a high percentage of moderate to vigorous PA > 150 min per week in the participants. However, this proportion was lower than that reported in our study (97.7%). Most research, including our work, has attempted to have an approach to measure and describe the intensity, frequency, and time spent on PA in the population through validated instruments, such as the International Physical Activity Questionnaire short form (IPAQ-short) and the Minnesota Leisure Time Physical Activity Questionnaire [55], except for Mieziene B et al. [57], who applied a two-question self-report which was not validated. Questionnaires and self-reports are definitely feasible and reliable tools which allow us to approach the PA of the population.

There is evidence that cigarette smoking produces a decrease of 5.7% in HDL-C concentrations [29]. A smoking habit was present in 10.3% of our total population without differences between the HDL-C groups. Similar results were obtained in young U.S. adults; about 14.1% aged 25–44 years and 7.4% aged 18–24 years were current smokers [58]. In contrast with our results, smoking prevalence was higher in Korean (23.3%) (18–24 years old) [59] and Italian (24%) [60] young adults (18–34 years old). These countries have a smoking prevalence that is twice as high as that observed in our case, which could be due to differences in the educational level, smoke-free campaigns, and legislation compliance in the countries [61,62]. In our study, the determination of smoking habits was conducted through generalized, structured questions, like the dichotomous question regarding smoking habits (yes/no) used by La Fauci V et al. [60]. Other works designed specific instruments (validation processes are not explicitly stated) to assess the prevalence of tobacco use among adults, such as the National Adult Tobacco Survey (NATS) questionnaire [58] and the National Health Interview Survey (NHIS) [59,63]. In all cases, it has been proposed to consider that smoking self-reports may underestimate the prevalence in adolescent and adult populations [64,65]. Moreover, it is important to consider that cigarette smoking has declined in recent years due to the increase in the use of e-cigarettes and marijuana in young adults across different continents [58,66,67,68,69]. Therefore, researchers should consider the use of other substances as a risk factor in this population.

Short-term low-to-moderate alcohol consumption has been associated with elevated HDL-C levels [30,70] through increased rates of apoA-I transport and, over time, with a slower decline in HDL-C [31]. However, long-term excessive alcohol consumption is associated with hypertriglyceridemia, which accelerates the catabolism of HDL-C [31,71,72]. The prevalence of current drinkers who are over 15 years old compared with other continents is higher (59.9% Europe, 54.1% America, 53.8% Western pacific) [73] than we observed in our work (34.1%) and similar (10.3%) to the reports of heavy alcohol use in U.S. young adults aged 18–25 (7.2%) [74].

Among dietary risk factors, a cross-sectional study in Canadian young adults found that a higher energy contribution from UPFs was associated with low HDL-C [75]. UPFs are often energy-dense and high in added sugars, trans fats, and sodium [34,76,77]. These components are related to an increase in hepatic de novo lipogenesis, reduced oxidation, and increased fatty acid uptake in tissues, which stimulate pathways that decrease HDL-C concentrations [32,35]. Although we would expect a lower proportion of UPF consumption in subjects with a healthy HDL-C concentration, the UPF intake in the whole population was very frequent and similar (both groups). Nearly half of the energy intake of the individuals came from PFs and UPFs, and almost a third of the energy intake came from UPFs (29.9%). These findings are in agreement with previous reports on the Mexican population aged >1 year old (29.8%) [78]. Compared with others countries, the energy contribution of UPFs in our population was lower that in than U.S. adults aged >20 years (55.5% of energy) [76], Canadian adults aged >18 years with metabolic syndrome (51.9% of energy) [75], UK children aged 3–8 years (56.8% of energy) [79], and Lebanese adults aged >18 years (36.5% of energy) [80]. These investigations have assessed UPF consumption through validated dietary data collection tools that were not specific and were not designed for the classification of the NOVA system, which is similar in our work. Future approaches for the association between metabolic indicators and UPF consumption should consider specific tools to estimate the NOVA classification system and include more details of processed foods in each group.

The percentage of energy from UPFs goes hand in hand with a poor diet that is energy-dense, with unbalanced macronutrients and a low micronutrient content [81], which was confirmed in our study according to the findings of the prevalence of nutritional adequacy. We found that one-third of the emerging adults have an intake above the recommended levels for energy (28.7%) and carbohydrates (29.5%), and close to half of that of total fatty acids (39.8%). Similar results were obtained in Brazilian young athletes (14–49 years). The study revealed that the estimated intake prevalence above the recommendation for lipids was 47.3% [82]. Almost all the participants had intakes above the recommendations of saturated fatty acids (SFAs) (87%). This proportion is similar to that previously shown in Mexican women (>20 years), about 82%, and higher than that reported in Mexican men (67%) [83]. The tools (reference guidelines and food tables) and software for analyzing energy and macronutrient intake used in these studies were relevant to their locality, e.g., the Brazilian nutritional guide for athletes [82] and the food composition database compiled by the National Institute of Public Health [83]. In the case of our study, we used the Institute of Medicine and the international guidelines for the prevention of dyslipidemia as energy and macronutrient references, as well as the SFFQ, validated for a Mexican population [39], and the Nutritionist Pro Diet software version 7.9, which use the United States Department of Agriculture (USDA) databases that contain the Mexican foods that are necessary for the dietary analysis of our population.

As expected in accordance with the dietary intake of the population, the ω6/ω3 ratio (9:1) exceeds recommendations, independently of HDL-C concentrations. This is consistent with previous findings in Lebanese (18–28 years) (10:1) [84] and Canadian young adults (20–29 years) (9:1) [85], and lower than the average reported in Western diets [86]. The optimal balance of the ω6/ω3 ratio (1/1 to 4/1) is essential for the prevention of chronic diseases [47]. Li et al. found in a meta-analysis that a low ω6/ω3 ratio could increase HDL-C concentrations [87].

Our study showed that emerging adults did not meet recommendations for key nutrients to promote a healthy lipid status, which could modify lipoprotein carriers [88], decrease HDL particle number and sizes [89], and increase apoB-containing lipoprotein particles [90]. To our knowledge, there are no previously published studies that have evaluated the adequacy of vitamins according to HDL-C concentrations. To our surprise, in subjects with healthy HDL-C and low HDL-C, the status of these micronutrients was not different (1–11 and 1–9 numbers of nutrients below the EAR, respectively).

For a long time, the nutrient adequacy percentage used the Recommended Dietary Allowance (RDA) as a reference value. However, the RDA cut-off point classifies subjects into insufficient or excessive intakes, without taking into account interindividual and intersubject differences. This would overestimate the needs of 97.5% of the population. For this reason, we assessed the prevalence of nutrient adequacy using the EAR cut-off point, which assumes that the same number of individuals who are misclassified as having an inadequate intake would be the same as those classified as having an adequate intake [42,91]. This use of the EAR has been implemented in other analyses [92]. Previous studies of nutrient inadequacy have been performed in a wide range of age populations [93,94]; hence, this comparison is biased.

None of the subjects met the recommendations for phytosterols. Among the vitamins, we observed a relatively high prevalence of inadequate intakes of vitamin E (100%), vitamin B7 (85.1%), vitamin K (58.2%), vitamin B5 (44.1%), and vitamin D (36.4%). Consistent with our results, other authors [93,95,96] found a high prevalence of inadequate vitamin E intake (84%) [93] in U.S. adults and vitamin D intake in Mexican, Greek, and North American adults (>20 years) [93,95,96]. The prevalence of inadequate vitamin K, B5, and B7 intake was not reported by these authors, so comparisons could not be made. The prevalence of vitamin D and vitamin E adequacy considers adequate intake (AI) as the dietary reference intake and cut-off point. In this sense, the proportion of the population with habitual intakes below the AI level can be inferred, but it cannot be concluded that such intakes do not meet their needs. This occurs because the requirement distribution is unknown and there is no EAR defined [42]. Although there was a percentage of excess energy from fats in the population, most of it came mainly from SFAs (red and processed meat and bakery products), which can be corroborated by observing that more than 80% of the population consumed SFAs above the suggested intake. However, the population had a lower than recommended intake of whole grains, vegetable oils, nuts, seeds, and fish, which could be the reason why Mexican emerging adults had an inadequate intake of vitamins D and E [97]. Furthermore, we only estimated the usual macro- and micronutrient intake from food; dietary supplements were not considered, so the adequacy prevalence could be underestimated. In addition, it should be emphasized that there are no previous studies that have evaluated insufficient vitamin intakes in the emerging adult population exclusively, so it is not possible to make comparisons of this variable in similar populations with a more realistic view. Research on the prevalence of vitamin inadequacy includes populations with very wide age ranges, including adults and older adults, who have different physical, social, emotional, and metabolic characteristics [3,4].

In addition, observations on the effect of vitamins and antioxidants (D, E, lycopene, beta-carotene) have mostly focused on cardiovascular risk prevention in general [98,99,100]. The specific effects of antioxidant intake or supplementation associated with HDL-C concentrations are still controversial and inconclusive [101,102]. Therefore, the effect of lifestyle on HDL-C concentrations needs to be further investigated, and the possible related molecular mechanisms should be elucidated in future research in emerging adults.

It is essential to consider that variations in HDL-C concentrations are a multifactorial trait [103,104,105]. While this research did not specifically address genetics, it is worth noting that previous studies have emphasized the significance of genetic factors and their interaction with the environment in relation to HDL-C concentrations [8,105]. Prospective twin studies have estimated that over 60% of the variability in HDL-C can be attributed to genetic factors [106]. Genomic approaches have identified more than 71 loci that influence HDL-C, with over 38 of them being associated exclusively with this lipoprotein [106,107]. The most implicated genes include *ABCA1*, *APOA1*, *APOE*, *LCAT*, *CETP*, *LIPG*, *LIPC*, *PLTP*, *LPL*, and *SCARB* [105,106], which play roles in HDL-C biogenesis and various steps of reverse cholesterol transport [106]. Therefore, the complexity of HDL-C concentration underscores the need for further research into the effects of lifestyle, potential related genetic factors, and their underlying molecular mechanisms, which remain not fully understood.

The findings of the present study should be interpreted in light of their strengths and limitations. The strengths of this study include the EAR cut-point method to assess the prevalence of inadequate nutrient intake in the same group, which theoretically would give the least biased prevalence of failure, and is even comparable with the results produced using the probabilistic method. Furthermore, the instruments for dietary collection were previously validated in the Mexican population and were applied by trained dietitians. It is the first study to analyze the dietary antioxidant intake in emerging adults, and we observed wide standard deviations, suggesting subjects with extreme antioxidant intakes (low or high). Nevertheless, limitations existed in this work. One of them is the cross-sectional nature of the study, which could not establish causality. Second, information bias is inherent in data collection tools.

## 5. Conclusions

Based on previously published associations between lifestyle factors and HDL cholesterol in adults in general, we found no differences between smoking, alcohol consumption, physical activity, and diet between the healthy HDL-C and low HDL-C groups in emerging adults. The prevalence of an inadequate intake of vitamins and omega-3 (EPA/DHA), and a high percentage of energy from FPs and UPFs, coupled with an excessive saturated fatty acid intake and ω6/ω3 ratio is of concern in this population. In this way, it is important to develop nutritional intervention programs that promote healthy eating habits throughout life through the early identification of risk factors and a better understanding of modifiable lifestyle factors in emerging adults. To verify these conclusions and their implications, further research on emerging adulthood is needed.

## Figures and Tables

**Table 1 nutrients-15-04568-t001:** Sociodemographic characteristics of the emerging adults according to HDL-C.

	Total	Healthy HDL-C	Low HDL-C	*p*-Value
N	261 (100)	191 (73.1)	70 (26.8)	
Age, years	22.7 ± 2.0	22.7 ± 2.0	22.7 ± 2.0	0.831
Women, n (%) Men, n (%)	188 (72) 73 (28)	130 (68.1) 61 (31.9)	58 (82.9) 12 (17.1)	**0.018 ***
Occupation Student, n (%) Professional, n (%)	181 (69.3) 80 (30.7)	132 (69.1) 59 (30.9)	49 (70) 21 (30)	0.890
Family history of Dyslipidemia, n (%)	68 (26.1)	143 (74.9)	50 (71.4)	0.575

Data are presented as mean ± standard deviation for continuous variables or *n* (%) for categorical variables. *p*-values resulted from analysis of independent samples (*t*-test for continuous variables and chi-square test for categorical variables). * *p* < 0.05 was considered as significant.

**Table 2 nutrients-15-04568-t002:** Lipid parameters of the emerging adults according to HDL-C.

	Total	Healthy HDL-C	Low HDL-C	*p*-Value
N	261 (100)	191 (73.1)	70 (26.8)	
Cholesterol, mg/dL	162.6 ± 32.0	166.5 ± 30.1	152.0 ± 34.6	**0.001 ***
HDL-C, mg/dL Women Men	53.5 ± 10.7 55.6 ± 10.5 48.1 ± 9.5	57.5 ± 9.2 60.7 ± 7.7 50.6 ± 8.4	42.6 ± 6.3 44.1 ± 5.9 35.5 ± 2.5	**<0.001 *** **0.012 *** **<0.001 ***
LDL-C, mg/dL	93.3 ± 25.0	93.6 ± 25.8	92.4 ± 22.9	0.733
Triglycerides, mg/dL	81.5 ± 44.8	79.6 ± 40.7	86.7 ± 54.4	0.260

Data are presented as mean ± standard deviation. *p*-values resulted from analysis of independent samples using a *t*-test. HDL-C, high-density-lipoprotein cholesterol; LDL-C, low-density-lipoprotein cholesterol. * *p* < 0.05 was considered as significant.

**Table 3 nutrients-15-04568-t003:** Lifestyle characteristics of the emerging adults according to HDL-C.

	Total (n = 261)	Healthy HDL-C (n = 191)	Low HDL-C (n = 70)	*p*-Value
**Lifestyle characteristics**				
Physical activity <150 min/week >150 min/week				
	6 (2.3)	4 (2.1)	2 (2.9)	0.510 ^a^
	255 (97.7)	187 (97.9)	68 (97.1)	
Smoking habits Non-smokers, n (%) Smokers, n (%) Ex-smokers n, (%)				
	223 (85.4)	161 (84.3)	62 (88.6)	0.388 ^b^
	27 (10.3)	21 (11.0)	6 (8.6)	
	11 (4.2)	9 (4.7)	2 (2.9)	
Alcohol consumption Adequate, n (%) Excessive, n (%)				
	234 (89.7)	170 (89.0)	64 (91.4)	0.653 ^b^
	27 (10.3)	21 (11.0)	6 (8.6)	
Alcohol consumption Non-drinker, n (%) Moderate intake, n (%) Excessive intake, n (%)				
	172 (65.9)	121 (63.4)	51 (72.9)	0.173 ^b^
	62 (23.8)	49 (25.7)	13 (18.6)	
	27 (10.3)	21 (11.0)	6 (8.6)	
Medications, n (%) Diabetes drugs	2 (0.8)	1 (0.5)	1 (0.5)	0.465 ^a^
Hormonal treatments (only women)	18 (6.9)	14 (7.3)	4 (5.7)	0.428 ^b^
Weight loss, n (%)	110 (42.1)	73 (38.2)	37 (52.9)	**0.034 ***
**Percentage of energy from NOVA food classification system**				
NP and MPF, % energy	49.7 ± 12.6	49.0 ± 12.8	51.4 ± 12.09	0.189
PCI, % energy	7.6 ± 3.8	7.5 ± 3.7	7.9 ± 4.1	0.452
PF, % energy	14.6 ± 7.1	14.7 ± 7.4	14.2 ± 6.3	0.603
UPF, % energy	29.9 ± 10.9	30.5 ± 11.5	28.1 ± 9.2	0.118
PF + UPF, % energy	44.5 ± 12.1	45.3 ± 12.4	42.4 ± 11.0	0.086

Data are presented as mean ± standard deviation for continuous variables or *n* (%) for categorical variables. *p*-values resulted from analysis of independent samples (*t*-test for continuous variables and chi-square test for categorical variables). ^a^ Analysis of variance comparing the values between groups was calculated based on the F-fisher ^b^
*p*-values resulting from Somers’ D analysis for ordinal variables. The weight loss was evaluated 3 months ago. NP, unprocessed; MPF, minimally processed foods; PCI, processed culinary ingredients; PF, processed foods; UPF, ultra-processed foods. * *p* < 0.05 was considered as significant.

**Table 4 nutrients-15-04568-t004:** Energy and macronutrient intake according to international recommendations for dyslipidemia in the emerging adults.

	Recommendation Reference	Total (n = 261)	Healthy HDL-C (n = 191)	Low HDL-C (n = 70)	*p*-Value
Energy, n (%)	IOM, 2006 [42]				0.842 ^a^
Inadequate		22 (8.4)	16 (8.4)	6 (8.6)	
Adequate		164 (62.8)	122 (63.9)	42 (60.0)	
Excessive		75 (28.7)	53 (27.7)	22 (31.4)	
%E from CHO, n (%)	EAS, 2021 [11]				0.698 ^a^
Below (<45%)		134 (51.3)	97 (50.8)	37 (52.9)	
Within (45–55%)		50 (19.2)	39 (20.4)	11 (15.7)	
Above (>55%)		77 (29.5)	55 (28.8)	22 (31.4)	
%E from FAs, n (%)	AACE, 2017 [10]				0.967 ^a^
Below (<25%)		60 (23.0)	43 (22.5)	17 (24.3)	
Within (25–35%)		97 (37.2)	71 (37.2)	26 (37.1)	
Above (>35%)		104 (39.8)	77 (40.3)	27 (38.6)	
%E from SFAs, n (%)	AACE, 2017, AHA 2013 [10,41]				0.715
Below (<7%)		34 (13.0)	24 (12.6)	10 (14.3)	
Above (>7%)		227 (87.0)	167 (87.4)	60 (85.7)	
%E from MUFAs, n (%)	AACE, 2017 [10]				0.279
Below or equal (≤20%)		252 (96.6)	183 (95.8)	69 (98.6)	
Above (>20%)		9 (3.4)	8 (4.2)	1 (1.4)	
%E from PUFAs, n (%)	AACE, 2017 [10]				0.874
Below or equal (≤10%)		226 (86.6)	165 (86.4)	61 (87.1)	
Above (>10%)		35 (13.4)	26 (13.6)	9 (12.9)	
%E from LA, ω6, n (%)	IOM, 2006 [42]				0.121 ^a^
Below (<5%)		101 (38.7)	81 (42.4)	20 (28.6)	
Within (5–10%)		152 (58.2)	104 (54.5)	48 (68.6)	
Above (>10%)		8 (3.1)	6 (3.1)	2 (2.9)	
%E from ALA, ω3, n (%)	IOM, 2006 [42]				0.489
Below (<0.6%)		125 (47.9)	89 (46.6)	36 (51.4)	
Within (0.6–1.2%)		136 (52.1)	102 (53.4)	34 (48.6)	
ω6/ω3 ratio	Simopoulos, 2016 [47]	8.5/1	8.4/1	8.7/1	0.535 ^b^
Within (2/1 to 4/1)		2 (0.8)	2 (1.0)	70 (100)	
Above (>4/1)		259 (99.2)	189 (99)	0	
EPA + DHA (g/d)	AHA, 2009 [48]				0.892
Below (<500 mg)		230 (88.1)	168 (88.0)	62 (88.6)	
Above or equal (≥500 mg)		31 (11.9)	23 (12.0)	8 (11.4)	
%E from TFAs, n (%)	AACE, 2017, AHA 2006 [10,41]	259 (99.2)	190 (99.5)	69 (98.6)	0.465 ^b^
Below (<1%)		2 (0.8)	1 (0.5)	1 (1.4)	
Above (>1%)					

All data are presented as number of persons (%) or mean ± standard deviation. *p*-values resulted from analysis of independent samples (*t*-test for continuous variables and chi-square test for categorical variables). ^a^
*p*-values resulted from Somers’ D analysis for ordinal variables. ^b^ Analysis of variance comparing the values between groups was calculated based on the F-fisher. %E; percentage of energy; FAs, fatty acids; SFAs, saturated fatty acids; MUFAs, monounsaturated fatty acids; PUFAs, polyunsaturated fatty acids; TFAs, trans fatty acids, EPA, eicosapentaenoic acid; DHA, docosahexaenoic acid; ALA, alpha linolenic acid, LA, linoleic acid.

**Table 5 nutrients-15-04568-t005:** Prevalence (%) of failing to meet the EAR for each vitamin according to HDL-C concentration in the emerging adults.

	Total	Healthy HDL-C	Low HDL-C	*p*-Value
Number of nutrients below the EAR	1–11	1–11	1–9	
**Prevalence (%) of failing to meet EAR**				
Vitamin A	27 (10.3)	23 (12.0)	4 (5.7)	0.137
Vitamin C	3 (1.1)	3 (1.6)	0	0.566 ^a^
Vitamin D	95 (36.4)	68 (35.6)	27 (38.6)	0.659
Vitamin E	261 (100)	191 (100)	70 (100)	NA
Vitamin K	152 (58.2)	112 (58.6)	40 (57.1)	0.828
Vitamin B1	9 (3.4)	8 (4.2)	1 (1.4)	0.279
Vitamin B2	8 (3.1)	7 (3.7)	1 (1.4)	0.353
Vitamin B3	2 (0.8)	2 (1.0)	0	1.0 ^a^
Vitamin B5	115 (44.1)	82 (42.9)	33 (47.1)	0.544
Vitamin B6	5 (1.9)	4 (2.1)	1 (1.4)	1.0 ^a^
Vitamin B7	222 (85.1)	165 (86.4)	57 (81.4)	0.319
Vitamin B9	59 (22.6)	42 (22)	17 (24.3)	0.694
Vitamin B12	6 (2.3)	4 (2.1)	2 (2.9)	0.661 ^a^

All values are presented as number of persons (%). *p*-values resulted from analysis of chi-square test for categorical variables. ^a^ Analysis of variance comparing the values between groups was calculated based on the F-fisher. EAR, estimated average requirement. NA, not applicable.

**Table 6 nutrients-15-04568-t006:** Phytosterol and carotenoid intake according to HDL-C concentrations in the emerging adults.

	Total (n = 261)	Healthy HDL-C (n = 191)	Low HDL-C (n = 70)	*p*-Value
**Phytosterols (mg)**	32.4 ± 20.1	33.0 ± 21.7	30.8 ± 14.8	0.424
**Carotenoids**				
Total carotenoids (mg)	13.8 ± 63.1	13.9 ± 64.1	13.8 ± 60.6	0.893
Lutein + Zeaxanthin (mcg)	1786.6 ± 807.6	1806.3 ± 824.2	1733.9 ± 764.8	0.524
β-carotene (mcg)	4721.9 ± 2689.8	4653.2 ± 2687.4	4905.1 ± 2707.2	0.585
α-carotene (mcg)	1393.2 ± 972.5	1369.3 ± 951.6	1457.5 ± 1031.0	0.521
β-cryptoxanthin (mcg)	324.9 ± 255.0	331.7 ± 236.7	306.0 ± 244.0	0.446
Lycopene (mcg)	4032.0 ± 2099.5	4104.6 ± 2147.5	3838.8 ± 1968.3	0.375

Data are presented as mean ± standard deviation for continuous variables or *n* (%) for categorical variables. *p*-values resulted from analysis of independent samples (*t*-test for continuous variables and chi-square test for categorical variables).

## Data Availability

The data presented in this study are available upon request from the corresponding author.

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
