# Peer review of "The Influence of Lifestyle on High-Density Lipoprotein Concentration among Mexican Emerging Adults"

_nutrients, 2023, doi:10.3390/nu15214568_

Round 1

Reviewer 1 Report

Comments and Suggestions for Authors

Summary of manuscript: This was a cross-sectional study that investigated lifestyle factors and HDL-C concentrations in emerging adults. It was demonstrated that lifestyle factors did not differ “according” to HDL-C measurements. It was concluded that emerging adults possess lifestyle-related risk factors that could contribute to metabolic disorders. Therefore, nutritional interventions are important at this stage of life.   

General comments: I carefully reviewed this manuscript. The authors provided a study with interesting results. There are structural issues with certain sentences. In addition, there are singular/plural, past/present tense, and spelling/grammar issues in the manuscript. However, I commend the authors for an in-depth analysis. I provided my specific comments below.    

Abstract

Line 28: Should Estimated Average Range be changed to Estimated Average Requirement?

Lines 33-34: Are all omega-3 fatty acids and phytosterols “essential” nutrients? Please explain.

Methods

Lines 84-86: This is an example of an incomplete sentence. Therefore, please revise this sentence. In addition, please revise the other grammar issues in the manuscript.

Line 106: Should alpha-linoleic acid be changed to linoleic acid?

Line 112: Please define BMI.

Results

Table 1: Are the numbers correct in the table? For example, there were 70 men; however, the HDL-C categories are over 70.

Table 4: Please check the “Within” categories, as it appears that some should be changed to “Below” or “Above”.

Table 5: Should “falling” be changed to “failing”?

Discussion

Line 285: Please change “spend” to “spent”. Please carefully review the manuscript for spelling and grammar issues.

I applaud the authors for writing a very detailed and in-depth Discussion section. It is very much appreciated!

Comments on the Quality of English Language

As previously mentioned, there are spelling and grammar issues. Therefore, please carefully review the manuscript. 

Author Response

Response to general comments: Thank you for all your comments, they were very valuable in order to improve the manuscript. Changes were applied to the document, and they can be identified in red font and yellow colored text.

Abstract

Comment 1: Line 28: Should Estimated Average Range be changed to Estimated Average Requirement?

Response 1: Definitely, the correct terminology is Estimated Average Requirement (EAR), because there is not a range but a specific reference for micronutrients according to age and sex [Line 30].

Comment 2: Lines 33-34: Are all omega-3 fatty acids and phytosterols “essential” nutrients? Please explain.

Response 2: An apology, maybe the sentence was not clear. We are referring to a key nutrient to promote health lipid status or maintain a healthy HDL-C. Omega-3 fatty acids are essential fats as the body cannot synthesize them and must get them from food. This is not the case for phytosterols, which are not an essential nutrients. Therefore, we consider that the appropriate term could be “key nutrients to promote healthy lipid status”, substituting essential since it does not apply to phytosterols [Line 36].

Methods

Comment 3: Lines 84-86: This is an example of an incomplete sentence. Therefore, please revise this sentence. In addition, please revise the other grammar issues in the manuscript.

Response 3: We appreciate this observation. We realized the lack of connection between the words in the manuscript [Lines 99-103]. The text can be now read as follows:

"Eligibility criteria included: men and women aged 18 to 30 years, university students in their last semesters of health sciences or recent graduates. In addition to clinical records with complete information on sociodemographic, anthropometric, clinical, biochemical data and a complete semiquantitative food frequency questionnaire (SFFQ) of the subjects."

Comment 4: Line 106: Should alpha-linoleic acid be changed to linoleic acid?

Response 4: Of course, thanks for pointing out the error, we had not noticed it in previous revisions. Totally agree, the correct term is linoleic acid. The alpha corresponds to linolenic acid, a type of omega 3 [Line 123].

Comment 5: Line 112: Please define BMI.

Response 5: Thank you for the recommendation. The definition of the abbreviation has been added to the manuscript [Line 128].

Results

Comment 6: Table 1: Are the numbers correct in the table? For example, there were 70 men; however, the HDL-C categories are over 70.

Response 6: We appreciate the detailed review. We performed the descriptive analyses again and there was indeed an error in the data (number and percentages) for males and females with respect to the total population [Table 1].

Comment 7: Table 4: Please check the “Within” categories, as it appears that some should be changed to “Below” or “Above”.

Response 7: Your comment was very appropriate. In the case of saturated and trans fatty acids, the categories "Within" were substituted "Below". However, in the case of monounsaturated and polyunsaturated fatty acids, it was adjusted as "Below or equal" to the suggested percentage and "Above or equal" to the suggested percentage in EPA+DHA [Table 4].

Comment 8: Table 5: Should “falling” be changed to “failing”?

Response 8: We appreciate all your comments on the precision use of words in English. The manuscript was modified [Table 5].

Discussion

Comment 9: Line 285: Please change “spend” to “spent”. Please carefully review the manuscript for spelling and grammar issues.

Response 9: Thank you very much. We will make a detailed revision of the grammar and writing in English to improve the manuscript. We have already made the change of "spend" to "spent" [Line 292].

Reviewer 2 Report

Comments and Suggestions for Authors

Please rewrite the abstract according to recommandations (aim, material, results, conclusion)

Please extend the introduction with relevant information to the aim of the study.

The conclusion: please shorten to essential findings of the study.

Author Response

We sincerely appreciate all the comments to transmit more clearly the information about our work. Changes were applied to the document and can be identified in red font and yellow colored text.

Comment 1: Please rewrite the abstract according to recommandations (aim, material, results, conclusion)

Response 1: We restructured the abstract in the manuscript such in a way that the distinction of each section was clearer [Lines 24-40].

The text can be now read as follows:

“Previous studies have highlighted the role of lifestyle on HDL-C concentrations in adults. To our knowledge, the health and nutritional status of emerging adults have been understudied. The present study aimed to explore the most important lifestyle factors, micronutrient intake adequa-cy and percentage of energy from food processing according to HDL-C concentrations in emerg-ing adults. In this context, a cross-sectional analysis was conducted on 261 Mexican emerging adults apparently healthy. Lifestyle factors were collected through a structured survey and the prevalence of micronutrient intake inadequacy was estimated using the Estimated Average Re-quirement cut-point method. The percentage of energy from ultra-processed foods was assessed using the NOVA system. HDL-C was determined by the enzymatic colorimetric method. Statis-tical analyses were conducted in SPSS. The results revealed that lifestyle factors do not differ ac-cording to HDL-C status. The participants showed a poor nutritional diet, energy-dense and mi-cronutrient inadequate. Nearly half of the energy comes from processed and ultra-processed foods. Most participants did not meet recommendations for key nutrients (ϖ3 fatty acids and phytosterols) that promote health lipid status. In conclusion, regardless of their HDL-C levels, emerging adults exhibited lifestyle-related risk factors. The persistence of these findings over time could contribute to development of metabolic disor-ders in the future. It is crucial to increase understanding and to develop effective nutritional in-terventions during this critical phase of life.”

Comment 2: Please extend the introduction with relevant information to the aim of the study.​​

​​Response 2: Thanks for the suggestion, it will help to contextualize and support the objective of the article.The introductory section has been modified in the manuscript [Lines 45-88].

It can be read as follow:

“Emerging adulthood (EA) is considered a transition between adolescence and adulthood, typically comprising ages between 18 and 30 years [1,2]. Until recent years, it was considered a stage of low health risk. However, the constant changes in social, family, and personal contexts and the lack of economic independence [3,4] could represent a critical period for the adoption of lifelong risky lifestyles and eating behaviors [5,6]. During this period, different metabolic alterations may develop and go unnoticed for a long time until the symptoms of pathologies are evident. Previous reports have shown the presence of low HDL-C in university students [7–9]. This lipoprotein plays an important role in the prevention of cardiovascular diseases and can be altered by different factors. Lifestyle changes and specific dietary components of the diet have been recognized in the modulation of HDL. The most relevant HDL-related components that are already known to affect HDL-C concentrations (5 to 30%) are physical activity (PA), weight loss, smoking cessation, moderate alcohol consumption, energy restriction, dietary lipids (monounsaturated fatty acids, polyunsaturated fatty acids, saturated fatty acids, trans fatty acids), carbohydrates and antioxidants intake [10–21]. PA is considered one the most important strategies with an effect on HDL-C levels. It was widely demonstrated that regular aerobic exercise at least 150 min/week or resistance exercise 15-20 min/week and even leisure time and occupational PA according to PA guidelines recommendation can increase HDL-C concentrations [19,22–26]. Weight loss resulting from exercise [27] or caloric restriction [28] could be a good method to contribute to raise in HDL-C [19]. Other lifestyle components such as cigarette smoking produce a decrease in HDL-C concentrations [29] and for its part moderate alcohol consumption has been associated with an increase in HDL-C [30,31] and a slower decline over time [31]. Regarding dietary components, low and very low car-bohydrates diets [20] as well as their replacement with mono or polyunsaturated fat have been associated with an increase in HDL-C [14,16,19,32]. In a meta-analysis, high monounsaturated fat diets compared with high carbohydrates diets had a significant HDL improvement [14]. Even when comparing low-carbohydrate diets with low-fat diets, it has a greater HDL-C profile [17]. In addition, recent evidence suggests the consumption of ultra-processed foods (UPF) has been related with nutritional poor diets and the de-velopment of chronic diseases [33,34]. UPF consumption has been associated with changes in lipid concentrations (increase of total cholesterol and LDL-C) in children [35]. It has been previously reported the unfavorable effects of trans fatty acids resulting from industrial processing on HDL-C levels [11]. Evidence of lifestyle influence on HDL-C and specifically in emerging adults is limited and unclear. Little attention has been paid to the health and nutritional status of emerging adults. As well, a limited number of studies have described in detail the health and nutrition of this population. To our knowledge, the existing evidence about the effect of lifestyle, UPF intake and adequacy nutrient intake according to recommendations for dyslipidemia and cardiovascular disease prevention on the modulation of HDL-C concentrations focuses on a broad age range that includes adults and older adults (18-80 years) or children, who have entirely different physical, biological, and behavioral characteristics. In this regard, our aim was to explore lifestyle factors such as smoking habits, alcohol consumption, physical activity and diet characteristics as prev-alence of nutrient adequacy and UPF consumption according to HDL-C concentrations among emerging adults.”

Comment 3: The conclusion: please shorten to essential findings of the study.

Response 3: Thank you very much for your reflection. We restructured the conclusion in such a way as to convey in a more forceful way the most important results that respond to our main objectives [Lines 429-438].

The text looks as follows:

“Based on previously published associations between lifestyle factors and HDL cholesterol in adults in general, we found no differences between smoking, alcohol consumption, physical activity, and diet between the healthy HDL-C and low HDL-C groups in emerging adults. The prevalence of inadequate intake of vitamins, omega-3 (EPA/DHA), and a high percentage of energy from FP and UPF coupled with an excessive saturated fat acid intake and w6/w3 ratio is of concern in this population. In this way, it is important to develop nutritional intervention programs that promote healthy eating habits throughout life by early identification of risk factors and a better understanding of modifiable lifestyle in emerging adults. To verify these conclusions and their implications, further research on emerging adulthood is needed.”

Reviewer 3 Report

Comments and Suggestions for Authors

The study was performed to access the most important lifestyle factors according to HDL-C concentrations, however it was a cross-sectional study that involved a small group of participants. There was only 70 subjects in the group with low HDL-C level. The results did not show any significant differences in lifestyle factors according to HDL-C concentrations.

Besides the authors wrote that “it has been proposed to consider that smoking self-reports may underestimate in adolescent and adult populations [56,57]”, what can be another problem in a small group of participants.

 It is also questionable to give average of HDL-C level for the groups with different amounts of men and women (Table 2).

Moreover the authors compared the vitamins intake between groups with healthy and low HDL-C level, but they did not discussed what is (or what may be) the association between vitamins and HDL-C level.

The results should be revised, it is doubtful than none of the participants did not reach the recommended value of vitamin E. Takin into account that there was an excessive percentage of energy from fats it is wondering that 100% of participants had vitamin E intake  below the AI. Usually there is a problem with vitamin D intake, not with vitamin E.

The authors wrote that “The results of the antioxidant intake showed that none of the participants reached the suggested daily intake of phytosterols of 2g per day”. I supposed that authors wrote here about phytosterols that have an impact on LDL-C level when are eating in the amount of 2-3g per day, but these compounds are not antioxidants and it is known that their content in food is low and therefore it is recommended to eat food fortified in phytosterols for people with dyslipidemia. Were there any questions in the SFFQ about products enriched with phytosterols?

Conclusions should be strictly associated with the aim of the study. Most of the conclusions should be in the discussion section. 

Author Response

We sincerely appreciate all your comments and suggestions to strengthen the paper. Changes were applied to the document and can be identified in red font and yellow colored text.

The answer to each observation or question is found in the attached file. Because some tables were designed with information to support some answers, we considered this space not the best way to present it.

Therefore, we appreciate the review of the attached file.

Thank you

Round 2

Reviewer 3 Report

Comments and Suggestions for Authors

Please add to the tekst the explanation that was given in "response 6":

Previous research in American adults have also shown a high prevalence of inadequacy for vitamin E (84%) (Reider C, 2020). Since the prevalence of vitamin D and vitamin E considers adequate intake (AI) as the dietary reference intake and cut-off point, only limited inferences can be made. The proportion of the population with habitual intakes below the AI can be inferred, but it cannot be concluded that intakes do not meet requirements. This occurs because the requirement distributions are unknown and there is no EAR defined (IOM, 2006). Although there was a percentage of excess energy from fats, most of it came mainly from saturated fatty acids (red and processed meat and bakery products) which can be corroborated by observing that more than 80% of the population consumed saturated fatty acids above the suggested intake. However, the population had a lower than recommended intake of whole grains, vegetable oils, nuts, seeds and fish, which could be the reason why Mexican emerging adults had an inadequate intake of vitamins D and E (USDA Guidelines). In addition, we could observe an intake below the recommendations of ALA (48%), EPA+DHA (88%) in most of the emerging adults. Furthermore, we only estimated the usual macro and micronutrient intake from food, dietary supplements were not considered so, adequacy prevalence could be underestimated [Lines 206-214].

Author Response

We appreciate your suggestions. Changes were applied to the document and can be identified in red font and yellow-colored text.

Comment 1: Please add to the text the explanation that was given in "response 6":
Previous research in American adults have also shown a high prevalence of inadequacy for vitamin E (84%) (Reider C, 2020). Since the prevalence of vitamin D and vitamin E considers adequate intake (AI) as the dietary reference intake and cut-off point, only limited inferences can be made. The proportion of the population with habitual intakes below the AI can be inferred, but it cannot be concluded that intakes do not meet requirements. This occurs because the requirement distributions are unknown and there is no EAR defined (IOM, 2006). Although there was a percentage of excess energy from fats, most of it came mainly from saturated fatty acids (red and processed meat and bakery products) which can be corroborated by observing that more than 80% of the population consumed saturated fatty acids above the suggested intake. However, the population had a lower than recommended intake of whole grains, vegetable oils, nuts, seeds and fish, which could be the reason why Mexican emerging adults had an inadequate intake of vitamins D and E (USDA Guidelines). In addition, we could observe an intake below the recommendations of ALA (48%), EPA+DHA (88%) in most of the emerging adults. Furthermore, we only estimated the usual macro and micronutrient intake from food, dietary supplements were not considered so, adequacy prevalence could be underestimated [Lines 206-214].

Response 1: Explanation about the high prevalence of habitual intakes below AI in vitamin E shown into the population. The text added in the manuscript shown as follows:

Consistent with our results, other authors [93,95,96] found a high prevalence of inadequate vitamin E intake (84%) [93] in US adults and vitamin D in Mexican, Greek and North American adults (>20 years) [93,95,96]. The prevalence of vitamin K, B5 and B7 inadequate intake was not reported by the authors, so comparisons could not be made. Prevalence of vitamin D and vitamin E adequacy considers adequate intake (AI) as dietary reference intake and cut-off point. In this sense, the proportion of the population with habitual intakes below the AI can be inferred, but it cannot be concluded that intakes do not meet needs. This occurs because the requirements distribution is unknown and there is no EAR defined (IOM,2006). Although there was a percentage of excess energy from fats in the population, most of it came mainly from SFA (red and processed meat and bakery products) which can be corroborated by observing that more than 80% of the population consumed SFA above the suggested intake. However, the population had a lower than recommended intake of whole grains, vegetable oils, nuts, seeds and fish, which could be the reason why Mexican emerging adults had an inadequate intake of vitamins D and E (USDA Guidelines). Furthermore, we only estimated the usual macro and micronutrient intake from food, dietary supplements were not considered so, adequacy prevalence could be underestimated" [Lines 396-413].
